# Working Hours, Job Burnout, and Subjective Well-Being of Hospital Administrators: An Empirical Study Based on China’s Tertiary Public Hospitals

**DOI:** 10.3390/ijerph18094539

**Published:** 2021-04-25

**Authors:** Zhihui Jia, Xiaotong Wen, Xiaohui Lin, Yixiang Lin, Xuyang Li, Guoqing Li, Zhaokang Yuan

**Affiliations:** 1Jiangxi Province Key Laboratory of Preventive Medicine, School of Public Health, Nanchang University, Nanchang 330006, China; 401437618004@email.ncu.edu.cn (Z.J.); 401437619002@email.ncu.edu.cn (X.L.); linyixiang338@163.com (Y.L.); 2School of Health Sciences, Global Health Institute, Wuhan University, 115 Donghu Road, Wuhan 430071, China; 2020103050014@whu.edu.cn (X.W.); xuyangl6613@126.com (X.L.); 3The First Affiliated Hospital of Nanchang University, Nanchang 330006, China

**Keywords:** hospital administrators, working hours, job burnout, subjective well-being, tertiary public hospitals

## Abstract

(1) Purpose: To analyze the role of job burnout in connection with working hours and subjective well-being (SWB) among hospital administrators in China’s tertiary public hospitals. (2) Methods: A multi-stage, stratified, cluster random sampling method was used to select 443 hospital administrators in six tertiary public hospitals for study. The data were collected and analyzed using the working hours measuring scale, Maslach burnout, and the subjective well-being schedule. Pearson correlation, structural equation model, and bootstrap tests were conducted to examine the association between job burnout, working hours, and SWB. (3) Results: Among the 443 respondents, 330 worked more than 8 h per day on average (76.2%), 81 had the longest continuous working time more than 16 h (18.7%), and 362 worked overtime on weekends (82.2%). The prevalence of job burnout in hospital administrators was 62.8%, among which, 59.8% have mild burnout and 3.00% have severe burnout. In the dimension of emotional exhaustion, depersonalization, and reduced personal achievement, the proportion of people in high burnout was 21.0% (91/433), 15.0% (65/433), and 45.3% (196/433), respectively. Job burnout has a mediating effect between working hours and SWB, which accounted for 95.5% of the total effect. (4) Conclusion: Plagued by long working hours and severe job burnout, the hospital administrators in China’s tertiary public hospitals may have low SWB. Working hours have a negative direct impact on job burnout and SWB, and an indirect impact on SWB through job burnout as a mediator. Targeted strategies should be taken to adjust working hours to promote the physical and mental health of hospital administrators.

## 1. Introduction

Overwork generally existed in the Asian-Pacific region; in particular, working overtime is an emerging occupational and public health issue in East Asian societies [1]. Commonly, medical workers work long hours across the globe, and China’s situation is equally severe [2]. In recent years, a broad area survey in Eastern, Central, and Western China revealed that 81.5% of medical staff in tertiary public hospitals worked long hours [3]. Previous studies have shown that working long hours can lead to suicide or sudden death, affecting quality of life and physical fitness. Both longitudinal and cross-sectional studies have shown that working long hours can also lead to psychological problems, such as depressive symptoms, self-reported prevalence of psychosis, and psychological distress [4,5]. Moreover, as a risk factor for fatigue, over-pressure, negative emotions, and continual long-term work could hurt the work itself. In the medical field, working long hours is a predictive factor of reduced productivity, increasing medical errors, high absenteeism, abuse of sickness absence, and even turnover intention [6].

Job burnout is a response to chronic exposure to workplace strains and is marked by feelings of emotional depletion, cynicism, and a sense that no matter the effort, there will be no progress in your work [7]. It has three characteristics: (1) the feeling of energy expenditure or exhaustion; (2) increasing the perceptual distance from work, or work-related negative emotions or feelings of cynicism; (3) reducing professional performance [7]. In addition to the negative impact on individual physical and mental health, burnout also affects organizational commitment, turnover intention, and job performance [8]. The work characteristics of extreme intensity, high stress, and long working hours of health professionals make them one of the most susceptible groups to generate job burnout [9]. Studies in Greece, Pakistan, Arabic countries, Ecuador, Malawi, and Kaunas indicated that job burnout among medical staff is quite a challenge for the health system, which has a direct or potential negative impact on medical institutions, health workers themselves, and patients [10,11,12,13,14,15]. High occupational burnout is also widespread among medical staff in China. A systematic review showed that the prevalence rate of job burnout in China’s medical field is between 66.5% and 76.9% [16]. According to a nationwide survey of 10,626 public health workers conducted by the National Natural Science Foundation of China, 41.0% of respondents felt highly emotionally exhausted, 37.0% were highly depersonalized, and 34.0% reported decreased personal accomplishment [17]. Hence, it is necessary to pay continuous attention to the job burnout of medical personnel in China.

People pay more attention to the pursuit of happiness in today’s society. The World Health Organization (WHO) has put well-being on the agenda as an indicator of social justice as part of the European “Health 2020” policy framework [18]. The reason why subjective well-being is the main focus is apparent. Firstly, SWB is part of the emerging literature in mental health, with greater research significance [19]. Secondly, SWB has been identified as relating to health problems. A study using global data has shown that SWB plays more critical roles in prolonging lifespans than objective factors such as economic and medical indicators [20]. A study from mainland China has found a correlation between SWB of healthcare workers and their physical, mental, and spiritual health. Subjective well-being must be regarded as a common goal of each medical institution [21]. Thirdly, impaired well-being is one of the most prominent causes of reduced job involvement and absenteeism from the workplace, which can subsequently shape service provision and client outcomes [22].

Studies in Sweden and Iran have identified long working hours as a significant cause of job burnout [23,24]. It was also confirmed again by an online survey of the American Society for Clinical Pathology (ASCP). The survey proved workload, such as quantity of tasks and cases, is a significant factor leading to job burnout [25]. Significantly, job burnout symptoms can be shown through depression and anxiety, and overwork has been proven to be a risk factor for these emotions. This indicated that working hours could impact job burnout directly [5]. More specifically, in the expected direction, the more employees work more hours per week, the more they frequently interact with recipients, have high caseloads, thus experiencing more burnout [26]. The theory could be applied to hospital administrators.

Focusing on the effect of work hours on SWB, most previous studies have suggested that long work hours have an adverse impact on SWB [27]. Adjusting working hours is an important measure suggested by the American National Institute for Occupational Safety and Health (NIOSH) to promote workers’ well-being [28]. It has been demonstrated in a cross-sectional study of working hours and well-being of health care employees in Finland and Germany [29]. Furthermore, a large number of studies have determined a negative correlation between job burnout and subjective well-being. A study focused on tertiary-grade class-A hospitals in southwestern China using multiple stratified regression analysis revealed that the emotional exhaustion, depersonalization, and reduced personal accomplishment dimensions of medical personnel all had significant effects on subjective well-being [30]. A study in Korea putting some relevant factors, including burnout, into the SEM model found that the relation of the direct or indirect impact of burnout on well-being is established. The results explained 68.3% of the total variance of well-being [31].

Maslach, Jackson, and Leiter proposed that the existence of specific job demands (e.g., workload) and the lack of specific job resources can predict job burnout. Then, they will lead to a variety of negative outcomes (e.g., physical illness, reduced happiness, separation, absenteeism, etc.) [32]. The job demand–resource (JD-R) model (developed by Bakker et al. [33] and Demerouti et al. [34]) has become a mainstream conceptual framework used in job burnout research. The model proposed that working conditions can be divided into two broad categories, job demands and job resources. Burnout follows two processes in the model. Firstly, long-term work demands, e.g., workload, time pressure, and role stress, lead to exhaustion. Secondly, a lack of job resources, e.g., opportunities, rewards, job security, job identity, and supervisor and colleagues’ support, lead to undesirable outcomes. Obviously, the energy-driven process is “job demands → burnout → negative effect”, whereas the motivation-driven process is “job resources → engagement → positive effect” [35]. Bakker et al. [33] found that, in the JD-R model, the two processes operate independently. Thus, based on Maslach’s theory and energy-driven methodology, we hypothesize a mediating effect of job burnout on working hours and SWB.

Although previous studies have highlighted the importance of working hours in understanding individuals’ job burnout or SWB [25,27], we have a different context. Most of the existing related research has been conducted in Western countries, and rarely have thorough studies been conducted in China, even in Asia [27]. China has the largest population in the world. With the opening of the two-child policy and the increasing aging people, the quantity of patients and medical demands in tertiary public hospitals has reached an unprecedented level [36]. According to the *China Health and Family Planning Statistical Yearbook 2019*, the number of hospital visits of tertiary hospitals in China was 1.854.787 million, increasing by 143.9% over the year 2018 [37]. Due to these visits, nonstandard work hours (e.g., working in the evening, night shifts, or working on weekends) and high burnout in China’s tertiary public hospitals have become common in recent years.

Unlike other medical personnel, hospital administrators demand comprehensive skills. It encompasses hospital planning and operational activities to ensure adequate numbers and quality of trained human resources, effective financial management, disaster preparedness, health management information system utilization, support services, biomedical engineering, transport, and waste management to mention a few [38]. Furthermore, the implementation of standardized professional training programs in China means workers must engage in training, assessments, and other inspection activities. In the affiliated medical college, they even engage in teaching and scientific research [39]. Not surprisingly, hospital administrators often work overtime. The Chinese government announced a series of health reforms in 2009 to improve access to public health facilities and reduce the individual medical burden. However, the policy focusing on improving patient services may overlook the welfare of hospital administrators, and increase their workload [40]. Researchers have confirmed that the lack of benefits, heavy workload, and deteriorating work environment would adversely affect the subjective well-being of hospital administrators, making their happiness index lower than medical technicians and doctors [41]. Therefore, it is meaningful to investigate the hospital administrators.

The aim of this study is to (1) report the results of measuring the levels of working hours, job burnout, and SWB among hospital administration; (2) to understand the relationship between aspects; and (3) to examine the mediating effect of job burnout on working hours and SWB.

## 2. Materials and Methods

### 2.1. Participants and Procedures

In June 2019, we surveyed among hospital administrators attending six tertiary public hospitals in Jiangxi province, and the hospitals were selected through stratified random cluster sampling. The specific sampling method is as follows: (1) the province is divided into northern, central, and southern regions according to geographical location. (2) One city is selected for each region based on the number and scale of tertiary public hospitals and the expert opinion. (3) Two hospitals were randomly selected from each city, including one general hospital and one specialized hospital. (4) All the on-duty workers engaged in hospital administration for more than 1 year were investigated. A total of 482 questionnaires were distributed. The response rate of the survey was 89.8%.

### 2.2. Measures

#### 2.2.1. Demographics

Socio-demographic information included sex, age, marital status, educational background, and hospital category. The hospital administrators were divided into four age groups, including 30 years old and younger, 31–40 years old, 41–50 years old, 51 years old and older. Moreover, marital status and educational background were married and not married, below bachelor’s degree and above, respectively. Hospitals were divided into general hospitals and specialized hospitals.

#### 2.2.2. Working Hours

We assessed working hours with an existing instrument, including working hours per day, continuous working hours, and overtime on weekends as the three items [42]. The factorial analysis results showed that the Kaiser–Meyer–Olkin measure of sampling adequacy was 0.811 (>0.700), and Bartlett’s test of sphericity was χ^2^= 167.05, *P* < 0.001, suggesting that it was more suitable for factorial analysis. The Cronbach α coefficient of the questionnaire was 0.71. In addition, for better description and comparison, we classified the responses of working hours according to the Labor Law of the People’s Republic of China and the actual situation of hospital administrators. Among them, the answers of average daily working hours were dichotomized into “≤8 h”, “>8 h”; continuous working hours were divided into “≤8 h”, “9–16 h”, “>16 h”, and overtime on weekends were “0 days” or “≤1 day”, “>1 day”, “2 days”.

#### 2.2.3. Job Burnout

We measured job burnout by adopting the international general Maslach Burnout Inventory Human Service Survey (MBIHSS) [32], which is the most widely used measurement tool in the study of job burnout. The MBI includes emotional exhaustion (9 items), depersonalization (5 items), and reduced personal accomplishment (8 items), with a total of 22 items. The hospital administrators’ feelings were elicited on a 7-point Likert scale: “never having those feelings” to “having those feelings everyday” were scored on a scale of 0–6. The Cronbach’s alphas for three subscales were 0.873, 0.734, and 0.859, respectively, which was considered acceptable in reliability [17].

Emotional exhaustion refers to the excessive consumption of individual emotional resources, exhaustion, and loss of energy. In this dimension, less than 19 points was defined as “low burnout”, 19–26 points as “moderate burnout”, and more than 26 points as “severe burnout”.

Depersonalization refers to an individual’s negative, indifferent, and excessively distant attitude toward the object of service. In this dimension, less than 6 points was defined as “low burnout”, 6–9 points as “moderate burnout”, and more than 9 points as “severe burnout”.

Reduced personal accomplishment refers to the decline of the individual’s sense of competence and job achievement. In this dimension, more than 39 points was defined as “low burnout”, 34–39 points as “moderate burnout”, and less than 34 points as “severe burnout”.

In order to better describe the burnout states, a weighted burnout score was introduced. The equation was defined as burnout = 0.4 × exhaustion + 0.3 × depersonalization + 0.3 × reduced personal accomplishment. The same strategy was adopted by Ahola et al. [43]. Higher scores indicated a higher level of job burnout. The extent of burnout scores of all hospital administrators was classified into 3 categories: no burnout (scores 0–1.49), mild burnout (scores 1.50–3.49), and severe burnout (3.50–6), and respondents with mild or severe burnout were defined as “burnout cases”.

#### 2.2.4. Subjective Well-Being (SWB)

The measuring instrument comes from the General Well-being Schedule (Fazio, 1977), which is a standard test developed for the National Center for Health Statistics (NCHS) to assess participants’ statements of happiness [44]. In 1996, a Chinese professor revised the scale. The Cronbach’s alpha of the revised scale was 0.77, the internal consistency coefficient was 0.91 for males and 0.95 for females, and the correlation between the score of a single item and the total was between 0.48 and 0.78. The scale adopts the 11-point scoring method, which is 0–10 points. The higher the total score, the higher the SWB [45].

### 2.3. Ethics Statement

This study was approved by the medical ethics committee of the first affiliated hospital of Nanchang University, and conducted with the informed written consent of the hospital. Participants were informed that their participation in the study was voluntary, and their responses were anonymous.

### 2.4. Data Analysis

EpiData 3.1 software (The Epi Data Association, Odense, Denmark) was used for data entry, and SPSS 22.0 and Amos 22.0 software were used for data analysis and processing. Descriptive statistics were used to describe the socio-demographic aspects and working hours. Comparison of the means of two groups was performed using an independent sample *t*-test. ANOVA was used for comparison of the means of multiple groups. The Pearson correlation test was used to analyze the correlations between aspects, and exploratory factor analysis was used to conduct a common method bias test on the data.

The structural equation model (SEM) can be used to estimate abstract concepts using measured variables, examine the complex causal relationships among variables using feedback loops, and improve the accuracy and credibility of model results by considering the influence of measurement error [46]. The model is suitable for identifying the mediating effects of variables and thus was used in this study.

In this study, working hours and job burnout are constructs that cannot be directly measured, so we set latent variables. Latent variables are composed by indicators built by the parceling method and each latent variable was composed by three parcels. The parceling method is more advisable since it reduces type I errors in item correlations, and it lessens the likelihood of a priori model misspecification [46].

A bootstrap method was used to test the mediation effect. The completely standardized effect size was estimated by 95% confidence intervals (CIs) with a bootstrap of 5000 times. A significant indirect effect was presumed when the 95% CIs did not contain 0 [47]. The significance level was set at 0.05 (two-tailed). 

## 3. Results

### 3.1. The Scores of Various Dimensions of Job Burnout among Hospital Administrators

In this study, 62.8% of hospital administrators have job burnout, among which, 59.8% have mild burnout and 3.0% have severe burnout. In the dimension of emotional exhaustion, depersonalization, and reduced personal achievement, the proportion of people in high burnout was 21.0% (91/433), 15.0% (65/433), and 45.3% (196/433), respectively, as shown in Table 1.

### 3.2. Demographic Distribution of Job Burnout and SWB among Hospital Administrators

More than half of the study participants were women (63.5%, N = 275). The age ranged from 21 to 67 years (mean 37.4). Most of the participants (81.7%, N = 371) obtained bachelor’s or higher degrees. In addition, 76.2% of hospital administrators worked >8 h per day on average, 18.7% had the longest continuous working time of >16 h, and 82.3% worked overtime on weekends. Among those, significant differences were observed in scores of job burnout and subjective well-being, regarding education level, average daily working hours, continuous working hours, and overtime on weekends (*P* < 0.01) (see Table 2 for details).

### 3.3. Correlation Analysis between SWB and Various Factors of Working Hours and Job Burnout in Hospital Administrators

The correlations between different dimensions of working hours, job burnout, and SWB are presented in Table 3. The results reveal that significant correlations exist between all three dimensions of working hours (*P* < 0.001), three dimensions of job burnout (*P* < 0.001), and SWB. Specifically, average daily working hours (*r* = −0.267, *P* < 0.001), continuous working hours (*r* = −0.536, *P* < 0.001), and overtime on weekends (*r* = −0.493, *P* < 0.001) were negatively correlated with SWB. Similarly, emotional exhaustion (*r* = −0.556, *P* < 0.001), depersonalization (*r* = −0.372, *P* < 0.001), and reduced personal achievement (*r* = −0.128, *P* < 0.001) were also negatively associated with SWB. The more likely each of these factors is to occur, the more likely these hospital administrators are to report lower subjective well-being.

### 3.4. Testing of Common Deviation Method

The Harman single factor test was used to test for common method bias. The results of unrotated principal component factor analysis showed that there were 11 factors with characteristic root values greater than 1, and the variation value explained by the first factor was 26.7% (less than the critical criterion of 40.0%), indicating that the common method of the study is not serious.

### 3.5. Mediating Effect of Job Burnout between Working Hours and SWB

The SEM is constructed and revised using Amos 21.0, to verify the following hypotheses: (1) Working hours in hospital administrators have a direct effect on job burnout and SWB; (2) Job burnout has a direct effect on SWB; (3) Working hours have an indirect effect on SWB through job burnout. The test results of the initial model (M1) showed that the fitting indices were not ideal. The model was revised according to the index modification indices, and the same strategy was adopted by Zito et al. [46] and Zhang et al. [48]. According to the principle of releasing one parameter at a time, the model was modified step by step, and two paths with a strong relationship were added (e5 → e6, e7 → e8). After adjustment, the model M2 fitting index met the fitting conditions, and the fitting was good, as shown in Table 4.

As shown in Figure 1, in Model M2, the path coefficients between working hours and job burnout (*β* = 0.783, *P* < 0.001), working hours and SWB (*β* = −0.889, *P* < 0.05), and job burnout and SWB (*β* = −0.033, *P* < 0.001) were significant after controlling for the influence of the basic characteristics of hospital administrators.

The direct and indirect effects in the model were tested using the bias-corrected non-parametric percentage bootstrap method. After repeated sampling 5000 times, the 95% confidence interval was calculated. Table 5 showed that the path coefficients of direct effect and total effect of working hours on SWB were −0.033 (*P* < 0.001, 95% CI: −0.039~−0.027) and −0.728 (*P* < 0.001, 95% CI: −0.779~−0.644), respectively. The indirect effect of working hours on SWB, “working hours → job burnout → SWB”, was −0.695 (*P* < 0.001, 95% CI: −0.764~−0.614). The results showed that 95% of the total effects and the mediating effects of job burnout did not include 0. In other words, the mediating effect of job burnout on the relationship between working hours and SWB of hospital administrators has been confirmed, and it is a partial mediating effect, with an explanatory result volume of 95.5%.

## 4. Discussion

This study focused on hospital administrators of tertiary public hospitals in China. It mainly revealed the status and relationship between working hours, job burnout, and subjective well-being (SWB) of hospital administrators. As hospital administrators, some work overtime on weekends frequently, so that more than half of them generate job burnout and reduced personal achievement in particular. Consistent with previous research, we found a significant direct impact for working hours on job burnout, working hours on SWB, and job burnout on SWB [49,50,51]. Moreover, the bootstrap test of this study showed that job burnout of hospital administrators plays a mediating role between working hours and subjective well-being.

China’s Labor Law stipulates that the maximum continuous working hours shall not exceed 11 h. Still, this survey shows that 18.7% of the staff’s maximum continuous working hours are more than 16 h, indicating that some hospital administrators’ working hours have exceeded the legal standard. Chinese researchers consider that a workday of >8 h warrants adjustment as it contributes to job burnout by reducing job satisfaction [40]. However, 76.22% of administrators work an average of 8 h or above per day in this survey, requiring attention.

This study shows that more than half of hospital administrators (62.8%) suffer from burnout syndrome, which is lower than the administrators in tertiary public hospitals in Beijing, China (82.3%) [52]. The medical level in Beijing takes the leading position in China, attracting patients from all over the country to come here for medical treatment. This increases the workload of its administrators, resulting in a higher rate of severe job burnout. Twenty-one percent of administrators in the study are in a state of severe emotional exhaustion. The career of hospital administrators is relatively extraordinary. They need to master both management and medical knowledge, and their work involves a wide range of aspects, and complex and diverse interpersonal relationships. This will soon lead to the excessive expenditure of work enthusiasm, patience, and other psychological resources, resulting in emotional exhaustion. Previous review research results showed that the prevalence of severe depersonalization was 43.6% among resident doctors and 68.0% among internists [53]. Still, our study found that the bulk of severe depersonalization is 25.6% among hospital administrators, lower than physicians. On the one hand, the coordination of doctor–patient relationships is one of the essential job aspects of hospital administrators. They need to be patient with patients, so the handling of the interpersonal relationship (mainly for doctor–patient relationships) is gentle. On the other hand, unlike doctors’ on-call and shift systems, hospital administrators have relatively regular working hours, so the degree of depersonalization burnout is relatively low. In this study, 50.6% of the participants showed severe low personal accomplishment, higher than previous studies in some Asian and African countries with PA rates ranging from 8.8% to 34.6% [13,53]. This may be because the main force of the hospital is doctors and nurses. However, administrators are in an affiliated, coordinated, non-central position, and the sense of value and significance of their work is low.

Significant gender or age differences in SWB have not been found in this study. A few studies found women reported an extremely high level of SWB [54]. Similarly, some researchers found a decline in SWB as people grow older, and others said that SWB follows a U-shape across age [55]. However, their conclusions were refuted by a survey in which no differences were found [56,57]. This suggests that the determination of relationship was determined to be unstable in cross-sectional studies. Other correlates may also be more important than gender or age, given that different studies have different control variables [58]. When more critical variables appear, the gender or age effect often disappears. The SWB difference between public hospital and specialized hospital administrators is not statistically significant, which is different from the findings of Lorber et al. [59]. The situation may be because the hospitals in this survey were all tertiary public hospitals with a high patient flow, causing the hospital administrators to be in a high-intensity working state. Thus, the difference in SWB among respondents was not noticeable. The difference in SWB is affected by the level of education. That is, the highly educated tend to report lower levels of subjective well-being than less educated persons. A higher education degree may urge employees to evoke higher expectations of promotion and remuneration. Education may interfere with SWB if it leads to expectations that cannot be met [60], and this explained the results of this study.

Numerous studies have shown that working long hours limits the opportunities for recovery necessary to be healthy and happy [6]. In addition, it has a detrimental effect on SWB through individuals’ fatigue, health deprivation, and time pressure [27]. Our study shows that a significant negative correlation exists between working hours and SWB, which proves this again. Long working hours is a risk factor for poor health behaviors or habits, such as causing unhealthy sleeping and dietary habits, lack of exercise and smoking, which can, in turn, lead to poor mental health [61]. According to effort–recovery theory, working long hours would lead to reducing effort investment and prolonging recovery time. Consequently, long hours of work can result in inadequate recovery of human function, poor mental health, and disruption of physiological processes, reducing well-being [62]. Moreover, previous studies have proved that working on the weekends generally reduces family and shared leisure time, resulting in a work–family conflict (WFC) increase, which negatively affects the SWB of individuals and families [63]. It is worth mentioning that a report in the United States that simultaneously studied the relationship between WFC and burnout and well-being showed that WFC had a more significant effect on burnout than on well-being [64]. However, in this study, 81.3% of respondents work overtime on weekends, which cannot be ignored.

The mediating effect model verifies the establishment of the path of working hours → job burnout → subjective well-being, indicating that the job burnout of hospital administrators can strengthen the emergence of a low level of subjective well-being due to working hours. Job burnout is a “prolonged response to chronic interpersonal stressors on the job” [65]. Contrary to acute stress, job burnout over time is relatively stable and essentially cumulative [32]. Hospital administrators in tertiary hospitals are in the working environment of coordinating doctor–patient relationships, maintaining the smooth operation of the hospital, and dealing with various complicated affairs. When the workload in unit time exceeds their abilities, they need to work overtime to complete the task, and most importantly, this long-term work status is continuous. Working long hours can lead to exhaustion if not effectively addressed, which is a harmful psychological syndrome that involves a chain reaction of physical, emotional, and mental states [15,17,66]. Additionally, when employees generate burnout such as irritability, negativity, lack of emotional engagement, and doubt about their value, their psychological and social functions would be negatively affected, affecting the evaluation of life satisfaction and reducing SWB. The job demand–resource model (JD-R model) believes that the manifestation of the main effect of job demands (mainly including workload and time pressure) is to trigger relatively independent psychological processes. This depletes workers’ physical and mental resources, resulting in job burnout and low levels of subjective well-being and other adverse results [34]. The mediation model of this study also fully confirms this principle.

### Strengths and Limitations

To our knowledge, this is the first study to report the working hours, job burnout, and SWB of hospital administrators systematically. It is also one of the few studies that explores the relationship between workers’ working hours, job burnout, and SWB comprehensively by constructing a SEM intermediary model. Our research takes hospital administrators as the objects and breaks through the design idea of focusing on doctors and nurses in the past, suggesting the relevant departments and hospital leaders pay enough attention to the physical and mental health of the hospital administrators.

The limitations of this study are as follows: First, the samples were selected from six hospitals of one province, so the extrapolation of conclusions at the national level could be challenged. One should pay attention to this background when summarizing and interpreting the research results. Second, it was a cross-sectional study, so the interpretation of causal inference for the results was limited. It would be of benefit to confirm the causality with longitudinal data in future studies. Third, the measurement of relevant indicators is from the respondents’ self-reports, so there is inevitably reaction bias and selection bias, which might have an impact on the results. Finally, it is monotonous that this study only takes working hours to factor in job burnout and subjective well-being. Future research can consider and discuss other psychological variables, such as occupational stress, work–family conflict, etc.

## 5. Conclusions

Many administrators in China’s tertiary public hospitals are working hours beyond legal control, and more than half of them suffer from job burnout, especially reduced personal achievement. Working hours not only have a negative direct effect on job burnout and SWB, but also have an indirect effect on SWB through job burnout as a mediator. The results of the above research indicate that hospital administrators should be given more attention by adopting corresponding measures to adjust their state of working hours, job burnout, and improve their SWB. Such strategies would be beneficial to the physical and mental health of hospital administrators, and further create good medical conditions for patients and improve their satisfaction, so as to promote the smooth running of public hospitals.

## Figures and Tables

**Figure 1 ijerph-18-04539-f001:**
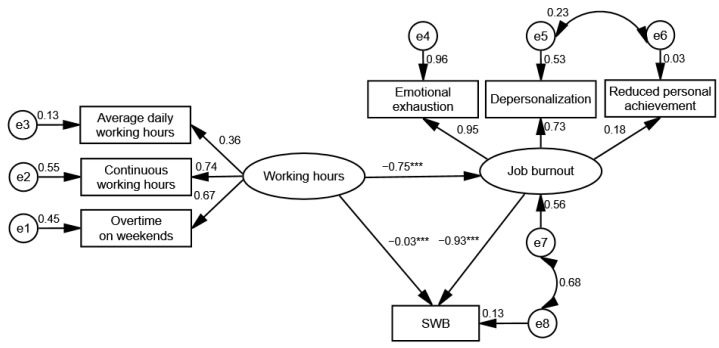
Mediating role of job burnout in working hours and SWB (M2). Note: *** *P* < 0.001.

**Table 1 ijerph-18-04539-t001:** Describing dimensions of job burnout in hospital administrators.

Level	M ± SD	Low	Moderate	High
EE (9 items, 0–54)	17.31 ± 11.58	233 (53.8)	109 (25.2)	91 (21.0)
DP (5 items, 0–30)	5.94 ± 5.71	269 (62.1)	99 (22.9)	65 (15.0)
PA (8 items, 0–48)	20.71 ± 10.19	121 (27.9)	116 (26.8)	196 (45.3)

Note: EE, emotional exhaustion; DE, depersonalization; PA, reduced personal accomplishment.

**Table 2 ijerph-18-04539-t002:** The distributions of job burnout and SWB among hospital administrators (N = 433).

Variables	Total	Job Burnout		SWB	
		± *s*	t/F-Value	± *s*	t/F-Value
Sex			0.539		0.157
Male	158 (36.5)	54.55 ± 16.03		42.68 ± 14.83	
Female	275 (63.5)	53.63 ± 17.72		42.47 ± 13.05	
Age (year)			0.840		1.403
≤30	123 (28.4)	54.67 ± 19.35		42.12 ± 13.94	
31–40	176 (40.7)	54.49 ± 16.01		41.9 ± 12.88	
41–50	84 (19.4)	53.94 ± 16.59		42.31 ± 13.1	
≥51	50 (11.6)	50.42 ± 15.88		46.26 ± 16.58	
Marital status			−0.124		−1.230
Married	344 (79.5)	53.76 ± 17.95		41.00 ± 15.50	
Not married	89 (20.6)	54.02 ± 16.1		43.00 ± 13.20	
Education			−3.209 **		2.222 *
Junior college and below	62 (14.3)	47.58 ± 18.15		46.11 ± 15.88	
Bachelor’s degree and above	371 (85.7)	55.03 ± 16.72		41.95 ± 13.24	
Hospital category			−0.009		0.546
Specialized	56 (12.9)	53.95 ± 15.01		43.48 ± 11.41	
General	377 (87.1)	53.97 ± 17.42		42.41 ± 14.03	
Average daily working hours (h)			−4.240 ***		5.155 ***
≤8	103 (23.8)	47.85 ± 19.44		49.09 ± 15.39	
>8	330 (76.2)	55.88 ± 15.87		40.51 ± 12.48	
Continuous working hours (h)			65.07 ***		96.58 ***
≤8	50 (11.6)	39.74 ± 18.50		60.98 ± 12.71	
9–16	302 (69.7)	52.28 ± 14.35		42.19 ± 10.90	
>16	81 (18.7)	69.04 ± 15.15		32.52 ± 12.45	
Overtime on weekends (d)			30.91 ***		49.86 ***
0	81 (18.7)	42.52 ± 16.58		55.49 ± 1375	
≤1	275 (63.5)	54.11 ± 15.22		41.30 ± 10.91	
>1	71 (16.4)	64.18 ± 14.95		34.15 ± 13.06	
2	6 (1.4)	81.17 ± 23.96		24.33 ± 10.38	

Note: * significant at the 0.05 level (two-tailed); ** significant at the 0.01 level (two-tailed); *** significant at the 0.001 level (two-tailed).

**Table 3 ijerph-18-04539-t003:** Correlation analysis between working hours, job burnout, and subjective well-being in hospital administrators.

Variable		*R*	*P* value
Working hours	Average daily working hours	−0.267	<0.001
	Continuous working hours	−0.536	<0.001
	Overtime on weekends	−0.493	<0.001
Job burnout	Emotional exhaustion	−0.556	<0.001
	Depersonalization	−0.372	<0.001
	Reduced personal achievement	−0.128	<0.001

**Table 4 ijerph-18-04539-t004:** Fitting index of the initial model (M1) and the adjusted structural equation model (M2).

Index	Suggested Values	Initial (M1)		Final (M2)	
		Measured Value	Evaluate	Measured Value	Evaluate
χ^2^/df	<3.00	8.537	Poor	2.521	Good
GFI	>0.90	0.928	Good	0.981	Good
RMR	<0.05	0.072	Poor	0.035	Good
RMSEA	<0.08	0.132	Poor	0.059	Good
AGFI	>0.90	0.855	Poor	0.955	Good
NFI	>0.90	0.888	Poor	0.972	Good
TLI	>0.90	0.849	Poor	0.969	Good
CFI	>0.90	0.899	Poor	0.983	Good

Note: GFI, goodness of fit index; RMR, root mean square residual; RMSEA, root mean square error of approximation; AGFI, adjusted goodness of fit index; NFI, normed fit index; TLI, Tucker–Lewis incremental fit; CFI, comparative fit index.

**Table 5 ijerph-18-04539-t005:** Bootstrap test of significance of mediation effect.

Item	Path	95% CI Value		Effect Size	Relative Effect Value
		Lower Limit	Upper Limit		
Direct effect	Working hours → SWB	−0.039	−0.027	−0.033	4.5%
Mesomeric effect	Working hours → Job burnout → SWB	−0.746	−0.614	−0.695	95.5%
Total effect		−0.779	−0.644	−0.728	

Note: CI = confidence interval.

## Data Availability

The data presented in this study are available on request from the corresponding author. The data are not publicly available due to requirements of investors.

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
