# Peer review of "Working Hours, Job Burnout, and Subjective Well-Being of Hospital Administrators: An Empirical Study Based on China’s Tertiary Public Hospitals"

_ijerph, 2021, doi:10.3390/ijerph18094539_

Round 1
Reviewer 1 Report
The text is cognitively and application-relevant. Part of the theoretical introduction with no structure defined by headings/paragraphs. It is worth considering keeping the structure because the introduction does not distinguish the part of the theoretical background and the research review. It is difficult to indicate the theoretical basis of the research (even in one of the paragraphs the phrase: "Maslach believed burnout is a response ...". It is worth considering Maslach's achievements rather than "believed".
The development of a methodology is to correctly, indicates ethical consent. The analysis of the results, carried out in a clear manner, contains the most important information.
The article discusses the obtained results and indicates the limitations of the research, but only briefly - there are also studies indicating the importance of job burnout and psychological factors (other than working time) that may be important both for SWB and burnout. It is worth noting that an analysis of selected factors was undertaken.
Numerous linguistic errors require rereading the work and supplementing; for example, the quotation at lines 379-380 has no footnote.
Author Response
To reviewer 1 #
Dear reviewer ,
Please find enclosed our revised manuscript entitled “Working hours, job burnout and subjective well-being of hospital administrators: An empirical study based on China's tertiary public hospitals (ijerph-1150455).” I very much appreciate the thorough and constructive criticism given by reviewers. I have made some changes according to the comments and suggestions of the reviewers. The text has been revised and edited for language so as to present a more clear and compelling narrative. These changes can improve the manuscript, I hope.
Below are the original review comments, and our responses are in blue text under each one.
1 |
The text is cognitively and application-relevant. Part of the theoretical introduction with no structure defined by headings/paragraphs. It is worth considering keeping the structure because the introduction does not distinguish the part of the theoretical background and the research review. |
Response: Thank you. We have accepted your suggestion to keep the structure. Paragraphs 1, 2 and 3 introduced the working hours, job burnout, and subjective well-being. Paragraph 4 and 5 explained the relationship between aspects. Paragraph 6 mainly stated the theoretical basis of this study. Next, paragraph 7 and 8 introduced current study context, and the work nature of the Hospital administrators, respectively. The last paragraph clarified the purpose of this study.The structure is explicit.Thank you.
2 |
It is difficult to indicate the theoretical basis of the research (even in one of the paragraphs the phrase: "Maslach believed burnout is a response ...". It is worth considering Maslach's achievements rather than "believed". |
Response: Thank you for your suggestion. In the first submitted manuscript, we only proposed a direct relationship between the observed variables on the basis of previous studies (paragraph 4 and 5 in the introduction). As you said, the theoretical basis of the research was not clear. In the revised draft, we made clear the theoretical basis of our research: Maslach’s theory and The job demand-resource (JD-R) model. The main explanations are in lines 112-127. Thank you.
Besides, In the original sentence, "Maslach believed burnout is a response ...", we made a change. Modified content considered Maslach's achievements rather than "believed". See lines 53-58, please.
3 |
The development of a methodology is to correctly, indicates ethical consent. The analysis of the results, carried out in a clear manner, contains the most important information. |
Response: Thank you very much for your affirmation. Thank you.
4 |
The article discusses the obtained results and indicates the limitations of the research, but only briefly - there are also studies indicating the importance of job burnout and psychological factors (other than working time) that may be important both for SWB and burnout. It is worth noting that an analysis of selected factors was undertaken. |
Response: Thank you for your professional advice. We only analyzed a selected factor (working hours) in our article, which is indeed one of our limitations. We are glad to adopt your suggestion and add the relevant content to the Limitations. check line 444-446, thank you.
5 |
Numerous linguistic errors require rereading the work and supplementing; for example, the quotation at lines 379-380 has no footnote. |
Response: Thank you for your professional advice. We checked the details of the article very carefully. And ask a English professional to check for us. We believe that the quality of the article will be greatly improved. For example, the footnote at lines 379-380 has been added. See lines 411-412. Thank you.
We thank for the carefully review again and have checked through the manuscript and made the needed changes and corrections in English writing.
Reviewer 2 Report
This study examines the relationship between working hours, job burnout and subjective well-being among employees in the administrative department of a hospital. A mediator model between these constructs is formed and tested with standard statistical methods. The authors have applied the methodology well, however, the text contains some shortcomings that limit the quality of the manuscript. The authors have applied the methodology properly, but the text contains some shortcomings that limit the quality of the manuscript.
I therefore recommend that the manuscript be checked for style and grammar by a native speaker. All other comments can be found in the attachment.

Author Response
To reviewer 2 #
Dear reviewer ,
Please find enclosed our revised manuscript entitled “Working hours, job burnout and subjective well-being of hospital administrators: An empirical study based on China's tertiary public hospitals (ijerph-1150455).” I very much appreciate the thorough and constructive criticism given by reviewers. I have made some changes according to the comments and suggestions of the reviewers. The text has been revised and edited for language so as to present a more clear and compelling narrative. These changes can improve the manuscript, I hope.
Below are the original review comments, and our responses are in blue text under each one.
1 |
This study examines the relationship between working hours, job burnout and subjective well-being among employees in the administrative department of a hospital. A mediator model between these constructs is formed and tested with standard statistical methods. The authors have applied the methodology well, however, the text contains some shortcomings that limit the quality of the manuscript. The authors have applied the methodology properly, but the text contains some shortcomings that limit the quality of the manuscript. I therefore recommend that the manuscript be checked for style and grammar by a native speaker. All other comments can be found below. |
Response: Thank you very much for your praise and professional suggestions. We have carefully modified it according to your opinions. Thank you for your advice, we ask the English professional to help us check the style and grammar, I believe the language of the article will be improved. In the revised version, each change is indicated in blue type. And it is marked on the left, for example, Q2 represents the second question you commented on. So it's convenient for you to do a quick review. Thank you.
2 |
In the abstract as well as in the main text: for percentages, please indicate only one decimal place (e.g. line 24: 18.7%). |
Response:Thank you for your professional suggestions. We have revised percentages to indicate one decimal place.You can find it in the full text. (e.g. line 24: 18.7%).Thank you.
3 |
In the abstract, main text, and tables, be sure to put spaces between words, numbers, and parentheses (e.g. average (76.22%)). |
Response: Thank you for your professional suggestions. We have revised the full text. Make sure there are spaces between words, numbers, and parentheses in the abstract, main text, and tables.s (e.g. line 23: average (76.22%)).Thank you.
4 |
25: Since this is a cross-sectional study, the term prevalence should be used instead of incidence. |
Response: Thank you for your professional suggestions. We have revised the full text. (e.g. Line 25: prevalence).Thank you.
5 |
25: Burnout results should be presented in a differentiated manner (give separate percentages for mild and severe burnout symptoms, see also line 184-186 |
Response: Thank you for your professional suggestions. We have revised it. See lines 25-26. Thank you.
6 |
44, 48, 55, 57: Pay attention to the correct comma character |
Response: Thank you for your professional suggestions. We have revised the full text. The revised part mainly focuses on the sentence: reduced productivity, increasing medical errors, high absenteeism, abuse of sickness absence. See line 51. Thank you.
7 |
48-49: Incorrect wording, please replace “increasing fault frequency” by “increasing medical errors”; “reducing productivity” by “reduced productivity”; remove the word “developing. |
Response: Thank you for your professional suggestions. We have revised this sentence. See lines 51-52. Thank you.
8 |
48: What is sick time abuse? |
Response: Thank you for your professional suggestions. The word of “sick time abuse”, means a person often takes sick leave when it is not necessary. We reviewed to the relevant literature , and used “abuse of sickness absence” instead of “sick time abuse”, which is more accurate. Check the revision at line 51 Thank you.
9 |
58-61: In the text, the authors give burnout data from Europe, but cite an American study (Shanafelt, 2012). Please provide correct source references. There are current figures from burnout research. If possible, more recent studies should be cited. |
Response: Thank you for your professional suggestions. We have checked and removed the inaccurate data from the article. Besides, in addition to the classical literature, most of the references in this paper have been cited more recent studies. Thank you.
10 |
60: Unnecessary text paragraphs (e.g. which situation is particularly grim) should be avoided. |
Response: Thank you for your professional suggestions. We have revised unnecessary text paragraphs. (e.g. Lines 67-68: you can not find “which situation is particularly grim”). Thank you.
11 |
61: Insert separator 10626 →10,626 |
Response: Thank you for your professional suggestions. We have revised. See line 69. Thank you.
12 |
61-63: Grammatically incorrect sentence: “According to a nationwide survey of 10,626 public health workers conducted by the National Natural Science Foundation of China, 41% of respondents felt highly emotionally exhausted, 37% were highly depersonalized, and 34% reported decreased personal accomplishment”. |
Response: Response: Thank you for your professional suggestions. We have revised it. See lines 68-72. Thank you.
13 |
66: workloads → workload |
Response: Thank you for your professional advice. We have revised.See line 90. Thank you.
14 |
66: The authors quote that many studies from Sweden and Iran have identified workload as a significant cause of job burnout but they don’t cite those studies. Please indicate the corresponding literature. |
Response: Thank you for your professional suggestions. We have added the corresponding literature. See line 89. Thank you.
15 |
67: Please remove: Meanwhile (incorrect wording in this context) |
Response: Thank you for your professional suggestions. We have revised. You won't find it on lines 88-90. Thank you.
16 |
69: Pay attention to correct citation form required by the Journal. This form is rather incorrect: Schaufeli W.B. and Shanafelt T. |
Response: Thank you for your professional suggestions. In accordance with the principle of quoting the latest literature, we have removed these two literatures. The sentence of “and this is confirmed again by the review of Schaufeli W.B. and Shanafelt T. [19,20]” in original manuscript will not be found in the modified version. Besides, we inserted a paragraph in place of the original sentence to better explain the information. See lines 94-97. Thank you.
17 |
73-74: Grammar: “People pay more attention to the pursuit of happiness in today's society”. |
Response: Thank you for your professional suggestions. We have revised. See line 74. Thank you.
18 |
74-75: What is meant by the term symbol? Social justice or something else? Please specify? Suggestion: “The World Health Organization (WHO) has put well-being on the agenda as an indicator of social justice as part of the European “Health 2020” policy framework”. |
Response: Thank you for your professional suggestions. We have revised. See lines 75-76. Thank you.
19 |
The sentences in this manuscript are often very long and convoluted. This makes it difficult to read and understand. Please make sure that the sentences are concise and meaningful. Please shorten nested sentences that are too long. e.g. Line 75-77; suggestion: “A study from Mainland China has found a correlation between subjective well-being of healthcare workers and their physical, mental and spiritual health. The subjective well-being must be regarded as a common goal of each medical institution.” e.g. Line 79-83: suggestion: Adjusting working hours is an important measure suggested by the American National Institute for Occupational Safety and Health (NIOSH) to promote worker’s well-being. It has been demonstrated in a cross-sectional study of working hours and well-being of health care employees in Finland and Germany Further examples, see line 94-97; 97-100: Please shorten the sentences and check the grammar; |
Response: Thank you for your professional suggestions. We have shorten the sentences and check the grammar. See lines 82-84; 99-103. Thank you.
20 |
84: remove “there is” |
Response: Thank you for your professional suggestions. We have removed. See lines 86-87. Thank you.
21 |
98: replace “them three” with the “aspects” |
Response: Thank you for your professional suggestions. We have revised. See line 162. Thank you.
22 |
92-94; same 97-100: The authors refer to few studies, but only one or none is cited. Provide sources. 101-109: Provide citation sources; |
Response: Thank you for your professional suggestions. In lines 92 to 94 of the original manuscript we have added sources. See line 131 in revised version. However, In the original manuscript line92-94; 101-109, we deleted it according to the suggestions of other reviewers. Therefore, no citation sources are provided in the revised version. Thank you.
23 |
109: Pay attention to correct writing of numbers 1.854.787; insert “over the year” 2018. |
Response: Thank you for your professional suggestions. We have revised. See lines 138-129. Thank you.
24 |
115-119: The sentence is too long and should be shortened. State a short and concise objective without specifying the region, clinic type, and method. This information belongs in the method section. |
Response: Thank you for your professional suggestions. The sentence has been shortened. State a short and concise objective without specifying the region, clinic type, and method. See lines 160-163. Thank you.
25 |
130-131: wrong wording: “The response rate of the survey was 89.8%; remove: “and 433 questionnaires were collected (redundant information). |
Response: Thank you for your professional suggestions. We have revised. See line 174. Thank you.
26 |
132-133: This information belongs to the result part. Please remove from the method part. Besides, provide mean ages in addition to min and max values. |
Response: Thank you for your professional suggestions. The information has been removed to the result part. Besides, we provided mean ages. See line 271. Thank you.
27 |
139-140: rewrite the sentence, use: married/ not married (adjust also in Table 2) |
Response: Thank you for your professional suggestions. We have rewritten. See line 181 (adjusted also in Table 2). Thank you.
28 |
143: The authors refer to an already existing instrument. Which instrument are we talking about? Please indicate sources |
Response: Thank you for your professional suggestions. The instrument is the working hours scale. We have indicated sources. See line 186. Thank you.
29 |
144-145: irrelevant information, please remove the sentence. |
Response: Thank you for your professional suggestions. We have removed the sentence that “Each entry was filled in by the respondents themselves, so we got continuous values”. So you can not find in line 186. Thank you.
30 |
156: grammar: “adopting” |
Response: Thank you for your professional suggestions. We have revised. See line 197. Thank you.
31 |
162: remove: (all >0.700) redundant information; line 163: remove: test |
Response: Thank you for your professional suggestions. We have removed. See lines 203-204. Thank you.
32 |
The table 1 belongs in the results section. Please pay attention to spaces between the numbers and brackets. Please rephrase the table title to "Describing dimensions of job burnout in hospital administrators." |
Response: Thank you for your professional suggestions. We have removed the table 1 to the results section. See line 236. Besides, we rephrased the table title. See line 267. Thank you.
33 |
180: correct citation (see Journal requirements) |
Response: Thank you for your professional suggestions. We have corrected the citation. See lines 561-562. Thank you.
34 |
184-186: sentence belongs to the result part |
Response: Thank you for your professional suggestions. The sentence has been removed to the result part. See lines 262-263. Thank you.
35 |
188: What do you mean by the “The scale root” ?; And here again: also watch out for spaces between words |
Response: Thank you for your professional suggestions. “The scale root”aims to explain the source of the scale. But through your suggestion, we realize that this is a vague expression. So we used “The measuring instrument comes from” instead of “The scale root in ”. See line 225. Besides, we have watched out for spaces between words. Thank you.
36 |
190-191: I’m wondering if this is a relevant information: (In 1996, Chinese Professor Duan Jianhua, a psychology professor at Peking University, revised the scale). Please be concise. |
Response: Thank you for your professional suggestions. This is a irrelevant information, and it is not necessary. So we revised it to “In 1996, A Chinese Professor revised the scale” in revised version. See line 227. Thank you.
37 |
198-201: The regularly performed requirements are important but they should not be explained in detail in such paper. As a reader of a scientific journal, I presuppose these pre-testing procedures. In my opinion the sentence can be deleted. |
Response: Thank you for your professional suggestions. The sentence has been deleted. You can not find it in “2.4. Data Analysis” in line 237.Thank you.
38 |
202: Use instead: Descriptive statistics were used to describe the socio-demographic aspects and working hours. |
Response: Thank you for your professional suggestions. We have revised. See lines 239-240. Thank you.
39 |
203-206: It is already reported in the text before that scales were formed. Please avoid redundant information. The test procedures used should be described as precisely as possible. Please pay attention to common formulations and styles in scientific journals |
Response: Thank you for your professional suggestions. We have revised. See lines 238-244. Paragraphs 3, 3 and 4 in the “2.4. Data Analysis” are supplemented according to the requirements of other reviewers. The test procedures used should be described as precisely as possible. And we paid attention to common formulations and styles in scientific journals. Thank you.
40 |
212-225: The paragraph does not provide any significant information that contributes to better understanding. In my opinion, this entire paragraph can be deleted. Especially since some formulations are chosen very unfortunate and therefore misleading. The essential information has already been presented in the Ethics Statement. |
Response: Thank you for your professional suggestions. We have deleted the entire paragraph. Thank you.
41 |
227-228: Please revise this sentence. What do you intend with this information and which method variables are meant? |
Response: Thank you for your professional suggestions. We looked at the literature, It is found that the sentence of "In this study, method variables were controlled during the survey by means of anonymous survey, diverse sample attributes, and reverse of some questions”is redundant. So we deleted the sentence. See the part of “3.2. Testing of Common Deviation Method” in line 295. Thank you.
42 |
241-242: If possible, please specify only the relative values and not both. |
Response: Thank you for your professional suggestions. We have revised, and specified only the relative values and not both. See lines 270-272. Thank you.
43 |
272-275: This sentence is incomprehensible. Please specify? |
Response: Thank you for your professional suggestions. After further verification, we found that this sentence is redundant information. So we revised it. See lines 307-309. After modification, the description is concise. Thank you.
44 |
275: Be sure to use consistent citations as prescribed by the journal |
Response: Thank you for your professional suggestions. We have revised and be sure to use consistent citations as prescribed by the journal. See lines 307-308 to confirm citations 46 and 48. Thank you.
45 |
In my opinion, comparisons with countries such as Norway, Palestine, or Ecuador are somewhat misleading because it involves different continents and health care systems. Here, comparison to studies from the Asian region and especially from China itself would be appropriate |
Response: Thank you for your professional suggestions. We have switched to the comparison with Beijing, China. See lines 354-357. Thank you.
46 |
321: Grammar: A verb is missing in this sentence |
Response: Thank you for your professional suggestions. We have revised. See lines 353-355. Thank you.
47 |
325-333: the comparison to health professionals is misleading, as their activities, responsibilities and schedules are different from administrative ones. 330-332: Check grammar and make sure to create short and concise sentences. |
Response: Thank you for your professional suggestions. We have deleted the comparison to health professionals. Besides, the sentences have been revised to be short and concise. See lines 357-362. Thank you.
48 |
349: check the citation |
Response: Thank you for your professional suggestions. We have checked and revised the citation. See line 394. Thank you.
49 |
344-351: The authors use sentences that are too long and convoluted. Some of the grammar is not correct. Please revise |
Response: Thank you for your professional suggestions. We have revised. See lines 377-381. Besides, we asked English Professionals to help us check the sentences in our article. We believe that the quality of the articles will be significantly improved. Thank you.
50 |
351-353: Why do the authors mention organizational support and leadership style in this context? These aspects were not covered in this survey. What do they refer to in their statement? |
Response: Thank you for your professional suggestions. organizational support and leadership style really weren't involved in our survey. We have removed this redundant information. See 387-390. Thank you.
51 |
359-361: Please provide literature sources for this statement |
Response: Thank you for your professional suggestions. We have provided. See line 397. Thank you.
We thank for the carefully review again and have checked through the manuscript and made the needed changes and corrections in English writing.

Reviewer 3 Report
The focus of burnout in different cultural context is helpful. It would help for authors to expand on this a bit more. What may be different in the current study context compared to prior ones?
What is closed management (line 113)?
Offering specific hypotheses or aims for the current study would be helpful. For example, it is clear that burnout is a key interest, but what about the relation to other variables? Authors should explain what is worth investigating here and reasons for doing so to help make a stronger introduction. For example, working hours is hinted as being important, but would help to describe specifically why it may impact burnout more directly. Why would subjective well-being be the main focus, etc.
Why use an SEM approach when a regression model would do? It seems like some additional work without sufficient justification for this methodology? Was it to test mediator effects? This could be explained further in the introduction. Why use latent variables when manifest variables could have been used? Use of these methods require some further explanation beyond typical regression modeling approaches.
Authors mention the focus is on hospital administrators. It seems like this would be a very small group but authors mention a very large sample from a small number of sites. It would help to describe/define the job roles and positions of the sample selected.
Why was working hours examined through factor analysis? This is unclear. It would help to describe the instrument/measure used. They also mention using specific cut points. This could be explained/clarified.
Analysis section could be presented just before the Results.
Section 2.5 could be presented in Results
Table 1 – may want to present this in the Results section.
It would help to describe the specific content/questions of the subjective well-being score further.
Points made in the Discussion about nature/scope of problem could fit better in introduction; such as describing the nature of the job and prior research with statistics, and discussing theories on how measures may relate to each other.
Line 420-422 – seem to be speculative rather than something the study tested directly
Minor phrasing revisions – several noted; may be more throughout the text. Editing review would be helpful.
Line 73 in the nowadays society
Line 98 between them three
Line 99 and it is also few have reported
Line 103 deep-water area and crucial area
Line 104 main landing points
Line 145 so we got continuous values
Line 190/191 – may not be necessary to cite author/affiliation
Line 198- write out p-p chart; abbreviation was unfamiliar
Line 272 As a result of the sample data has been scientifically tested and the 272 reliability of the questionnaire is very good
Line 305 them work overload and work overtime
Line 339 participants performed severe reduced personal accomplishment
Line 351 causing the hospital administrators are all in a high-intensity working state
Author Response
To reviewer 3 #
Dear reviewer ,
Please find enclosed our revised manuscript entitled “Working hours, job burnout and subjective well-being of hospital administrators: An empirical study based on China's tertiary public hospitals (ijerph-1150455).” I very much appreciate the thorough and constructive criticism given by reviewers. I have made some changes according to the comments and suggestions of the reviewers. The text has been revised and edited for language so as to present a more clear and compelling narrative. These changes can improve the manuscript, I hope.
Below are the original review comments, and our responses are in blue text under each one.
1 |
The focus of burnout in different cultural context is helpful. It would help for authors to expand on this a bit more. What may be different in the current study context compared to prior ones? |
Response: Thank you for your professional suggestions. We add a lot of content in the Introduction. The statement on the study context mostly on paragraphs 7-8. In these two paragraphs, we describe the current study context using the sentence of “we have a different context”, “rarely thorough studies have been conducted in China, even in Asia”, “China has the largest population in the world”, and “hospital administrators demand comprehensive skills”, etc. This is different from previous studies. See lines 129-134; 140-152. Thank you.
2 |
Offering specific hypotheses or aims for the current study would be helpful. For example, it is clear that burnout is a key interest, but what about the relation to other variables? Authors should explain what is worth investigating here and reasons for doing so to help make a stronger introduction. For example, working hours is hinted as being important, but you would help to describe specifically why it may impact burnout more directly. Why would subjective well-being be the main focus, etc. |
Response: Thank you for your professional suggestions. Firstly, we added the specific aims for this study. See lines 157-160. Thank you.
Secondly, the relationship between job burnout and working hours, as well as job burnout and subjective well-being is stated in paragraph 4 and 5 of the introduction. See lines 90-91, 103-105. Next, paragraph 6 explained a mediating effect of job burnout on working hours and SWB. See lines 112-127. Thank you.
Thirdly, we explained why working hours is hinted as being important, and described why working hours impact burnout directly. See lines 90-97. Thank you.
Finally, we added a lot in paragraph 3 to explain why subjective well-being is the focus. See lines 77-87. Thank you.
3 |
Why use an SEM approach when a regression model would do? It seems like some additional work without sufficient justification for this methodology? Was it to test mediator effects? This could be explained further in the introduction. Why use latent variables when manifest variables could have been used? Use of these methods require some further explanation beyond typical regression modeling approaches. |
Response: Thank you for your professional suggestions. We have made some further explanation in the article. “Why use an SEM approach” can be found in lines 241-245; “Why use latent variables” can be found in lines 246-251. Thank you.
4 |
Authors mention the focus is on hospital administrators. It seems like this would be a very small group but authors mention a very large sample from a small number of sites. It would help to describe/define the job roles and positions of the sample selected. |
Response: Thank you very much for your affirmation. Thank you.
5 |
Why was working hours examined through factor analysis? This is unclear. It would help to describe the instrument/measure used. They also mention using specific cut points. This could be explained /clarified. |
Response: Thank you for your professional suggestions. As you said, the reason we did factor analysis is to describe the instrument/measure. Thank you.
6 |
What is closed management (line 113)? |
Response: Thank you for your professional suggestions. “Closed management” refers to a management mode in Chinese hospitals. It does not adopt external management agencies, but only manages within the hospital itself. We have deleted this unnecessary statement in the article. Thank you.
7 |
Analysis section could be presented just before the Results. Section 2.5 could be presented in Results Table 1 – may want to present this in the Results section. |
Response: Thank you for your professional suggestions. We've adjusted the order.
Analysis section has been presented just before the Results. See line 233.
Section 2.5 has been presented in Results. See line 292.
Table 1has been presented in the Results section. See line 257.
Thank you.
8 |
Points made in the Discussion about nature/scope of problem could fit better in introduction; such as describing the nature of the job and prior research with statistics, and discussing theories on how measures may relate to each other. |
Response: Thank you for your professional suggestions. We have added some content to the Introduction. The nature of the job and prior research with statistics were described in lines 140-148. The theories on how measures may relate to each other were discussed in lines 112-127. Thank you.
9 |
Line 420-422 – seem to be speculative rather than something the study tested directly |
Response: Thank you for your professional suggestions. We have deleted original statement of “which is even more serious than the previous statistics for doctors” in the conclusion. The rest of the conclusion was tested directly. See line 449-453. Thank you.
10 |
Minor phrasing revisions – several noted; may be more throughout the text. Editing review would be helpful. 1) Line 73 in the nowadays society 2) Line 98 between them three 3) Line 99 and it is also few have reported 4) Line 103 deep-water area and crucial area 5) Line 104 main landing points 6) Line 145 so we got continuous values 7) Line 190/191 – may not be necessary to cite author/affiliation 8) Line 198- write out p-p chart; abbreviation was unfamiliar 9) Line 272 As a result of the sample data has been scientifically tested and the 272 reliability of the questionnaire is very good 10) Line 305 them work overload and work overtime 11) Line 339 participants performed severe reduced personal accomplishment 12) Line 351 causing the hospital administrators are all in a high-intensity working state |
Response: Thank you for your professional suggestions. We have made Minor phrasing revisions.
- "In the Nowadays Society" has been revised to "in today's Society". See line 74. Thank you.
- "Between them three" has been revised to "between aspects". See line 159. Thank you.
- “And it is also few have reported”has been deleted in our revised vision. Thank you.
- “Deep-water area and crucial area”has been deleted in our revised vision. Thank you.
- “Main landing points”has been deleted in our revised vision. Thank you.
- “So we got continuous values”has been deleted in our revised vision. Thank you.
- The sentence of “In 1996, Chinese Professor Duan Jianhua, a psychology professor at Peking University, revised the scale”has been revised. See line 223. Thank you.
- P-P chart is simple and practical, and can be used to quickly verify the distribution of data.His working principle was introduced by Reiss [1]. However, referring to suggestions from other reviewers, this is redundant content. So we removed it.
- The sentence of “As a result of the sample data has been scientifically tested and the 272 reliability of the questionnaire is very good”is considered redundant information. It has been revised. After modification, the description is concise. See lines 303-304. Thank you.
- "Them work overload and work overtime" has been revised to "some work overtime". See line 336. Thank you.
- "Participants performed severe reduced personal accomplishment" has been revised to "participants showed severe low personal accomplishment". See line 368.Thank you.
- “Causing the hospital administrators are all in a high-intensity working state”has been revised to "causing the hospital administrators in a high-intensity working state". See line 384. Thank you.
We thank for the carefully review again and have checked through the manuscript and made the needed changes and corrections in English writing.
- Reiss, R.D.; Thomas, M. Statistical analysis of extreme values: With applications to insurance, finance, hydrology and other fields. Statistical Analysis of Extreme Values: with Applications to Insurance, Finance, Hydrology and Other Fields: 2007.
Round 2
Reviewer 3 Report
The authors appear to have addressed all main comments from reviewers in detail.